# Decline of physical activity in early adolescence: A 3-year cohort study

**Vedrana Sember**, **Gregor Jurak, Marjeta Kovač, Saša Đurić, Gregor Starc***

Laboratory for the Diagnostic of Somatic and Motor Development, Faculty of Sport, University of Ljubljana, Ljubljana, Slovenia

* gregor.starc@fsp.uni-lj.si

## Abstract

This study analyses the changes of moderate-to-vigorous physical activity (MVPA) in a cohort of boys and girls aged 11 (n = 50) and 14 (n = 50). Physical activity was assessed with Bodymedia SenseWear Pro Armband monitor for 6 days in October 2013 and October 2016, considering 90% of daily wear time (21h and 40min). The initial sample (n = 160) included the children who wore the monitors at age 11 but the final analyzed sample included only those children from the initial sample (n = 50), whose data fulfilled the inclusion criteria at age 11 and 14. Physical fitness and somatic characteristics of the final sample (n = 50) were compared to a representative sample of Slovenian schoolchildren at ages 11 (n = 385) and 14 (n = 236) to detect possible bias. Changes in MVPA were controlled for maturity using the timing of adolescent growth spurt as its indicator. The average MVPA decreased more than one quarter (34.96 min) from age 11 to age 14. Children were significantly more active at age 11 than at age 14 (p < 0.01, d = 0.39). The timing of puberty onset in girls was significantly earlier (12.01 ± 1.0 years) (p < 0.01) than in boys (13.2 ± 0.75 years) (p < 0.01, d = 1.35). There was a significant gender difference in moderate-to vigorous physical activity at age 14 (p < 0.05, $\eta^2$ = 0.12) and between moderate-to vigorous physical activity at age 11 and 14 ($\eta^2$ = 0.11). After controlling for the timing of adolescent growth spurt the girls at age 11 showed significantly higher level of physical activity than at age 14 (p < 0.01, $\eta^2$ = 0.17). Early adolescence is crucial for the development of physical activity behaviours, which is especially pronounced in girls. The significant decline of MVPA between ages 11 and 14 in Slovenia are likely influenced by environmental changes since the timing of adolescent growth spurt did not prove as a factor underlying the decline of MVPA.

## Introduction

Physical activity (PA) is one of the key health determinants in life [1] and is one of the sources of total energy expenditure, which incorporates active energy expenditure, metabolism in resting state, the thermal effects of food digestion and body growth in children and adolescents [2]. Existing evidence shows that PA has positive effects on psychosocial health, and the functional capacity and wellbeing of people [3], whereas physical inactivity increases health risks [4,5]. Research findings indicate that PA in school-age, and moderate-to-vigorous physical

**Data Availability Statement:** The data underlying the results presented in the study are included in Supporting information in the xlsx format.

**Funding:** This research was funded by the Slovenian Research Agency within the Research

programme P5-0142 Bio-psycho-social context of kinesiology. The study received invaluable assistance in sports equipment, which was donated to the participating schools from the Olympic Committee of Slovenia and Elan Inventa, a sporting manufacturer and supplier.

**Competing interests:** The authors have declared that no competing interests exist.

activity (MVPA) in particular, positively influence the public health of the adult population, which resulted in the development of recommendations for policymakers to increase PA of the European population [6]. The first Slovenian PA guidelines for children are based on the World Health Organization (WHO) recommendations of at least 60 minutes of MVPA daily [7]. In order to reach the recommendations, children should undertake various types of PA (regular physical education (PE) in school, organized sport practice, active play, active transportation etc.). The guidelines also define the structure of suitable PA (e.g. warm-up, gradual effort, cool-down, and relaxation), as well as suggest appropriate types (exercises improving aerobic and muscular fitness) and frequency of exercise for children and youth [7,8]. The most elaborated national PA guidelines and WHO recommendations are suggesting for PA volume to decrease with age, which is rather unusual since environmental demands for sedentariness actually increase with age from childhood to adolescence and adulthood [9,10].

In both genders, MVPA levels decrease with age [11], especially during adolescence [12]. Most of the existing studies has found the decline of MVPA in children and adolescents [9,10,13–19], but did not take into account the differences in maturity. The decrease of MVPA may extend into adulthood [12,20,21] and seems to be more associated with biological age than with chronological age [22]. In this regard, we aimed to analyse the changes in habitual PA in a group of children in predominantly pre-adolescent period (at age 11) and predominantly early adolescent period (at age 14) also in relation to maturity. Children and adolescents from different countries experience diverse rhythms of everyday life due to different educational systems, social settings and cultural differences [23]. Considering this, it is important to analyse each national setting separately and adjust national recommendations to the actual needs and possibilities of local populations as well as to the most intensive periods of PA behavioural change [13,24–26], which is also reflected in behavioural patterns of PA in adulthood [27–29].

In addition to the described issues with PA recommendations, also the assessment of PA continues to present a huge research challenge due to multiple research approaches and tools that very often produce incomparable data. In the past, PA of Slovenian children and youth has been assessed predominantly with the use of questionnaires [30–33], but in recent years the development of various wearables [34–38] has enabled the utilization of more objective methods for estimation of PA. Wearable monitors, such as pedometers, load transducers, accelerometers, heart rate (HR) monitors, combined accelerometer/HR monitors and multiple sensor systems [34–36,39], can provide more accurate assessment of mechanical and physiological parameters of PA compared to self-report tools [35]. The most advanced generation of wearable monitors are multisensory devices such as SenseWear Armband Pro (SWA, Body-Media, Inc., Pittsburgh, PA), used in this study, which provides estimation of EE based on biaxial accelerometer, galvanic skin response and body heat loss [39,40]. SWA has been validated in children [41–43] and adults [44] as well of different intensities of PA [45–47].

Previous studies and comparisons on objectively and subjectively assessed data showed that Slovenian school children and adolescents (6–17 y) are among the most physically active in the world [48–52], with more than 80% reaching the 60 minutes of daily MVPA [49,50]. However, most of the comparisons focused on the preadolescent children [49,50,53,54]. The purpose of this longitudinal study is to analyse the changes in the amount of PA of boys and girls in the early stages of adolescence based on objectively assessed PA data. The monitored period between ages 11 and 14 is characterised by the onset of puberty and intensive somatic development and growth, resulting in increased developmental and behavioural heterogeneity among children within the same chronological age group [12,55]. Namely, differences in PA between pre-adolescent girls and boys can be a result of girls' typical earlier onset of puberty [56], while at the same time the individual differences in the onset of puberty within boys and girls can

very often exceed the mentioned sex differences. We hypothesised that MVPA decreases from age 11 to 14 in boys and girls, that decrease is more pronounced in girls and that maturity negatively affects this decrease.

## Methods

### Sample

Data for the present study were acquired from a longitudinal cross-sectional study *The Analysis of Children's Development in Slovenia* (ACDSi) which has been monitoring physical and motor development of Slovenian children and youth [57,58] and was being carried out every 10 years in the last five decades. The study uses a sentinel approach to provide a nationally representative sample of children from 11 different locations, stratified according to the type of environment (rural, rural-industrial, urban-industrial, urban). The primary sampling unit was school and the secondary one class. In the 2013 ACDSi our sample included 3,476 children and early-adolescents in the 6 to 14 age group. For the purpose of the present study only 11-year-olds (born on October 1, 2002, ± 6 months) of both sexes, who mostly did not yet experience adolescent growth spurt (pre-adolescents), attending primary school Grade 6 at the time of measuring, were included. The total sample of 11-year-olds comprised of 385 children, within which the initial sample of 160 children was invited to wear the SenseWear Pro Armband (SWA). The sample size was dictated by the number of available SWA sensors (n = 50) that still allowed us to monitor their physical activity of children from different localities within one month of the same season and therefore minimise the effects of weather conditions on children's habitual physical activity. The same schoolchildren wearing SWA at age 11 were measured again at age 14 when the majority already experienced the adolescent growth spurt and were thus considered as early-adolescents. The number of children from the initial sample with sufficient wear time on the required days at age 11 was 141 (boys = 70; girls = 71). The excluded 19 children did not accumulate sufficient wear time due to device malfunction or irregular wearing of the device. The final sample of children at ages 11 and 14 that met the inclusion criteria for both years was 50 (boys = 21). This final sample was included in the final analysis (Fig 1). The average age of the children included in the first round of the PA monitoring at age 11, was 11.3 ± 0.3 years (boys 11.4 ± 0.4; girls 11.3 ± 0.3) and the average age of the children in the second round at age 14 was 14.3 ± 0.3 years (boys 14.3 ± 0.3; girls 14.2 ± 0.3). Due to considerable exclusion and some drop-out from the first to the second round of the PA monitoring, we needed to identify a possible exclusion/drop-out bias (see S1 to S4 Tables) and compared the somatic characteristics and physical fitness of the children in the final sample to the initial sample (see flowchart, Fig 1). In order to investigate possible bias between initial sample and the total sample at age 11 and between the final sample and the non-included children from the initial sample at age 14, we compared their somatic characteristics as well as their physical fitness with the following variables: standing broad jump, obstacle course backwards, 20-s drumming test, flamingo balance test, sit and reach, shoulder circumduction, hand grip strength, bent arm hang, 20-m shuttle-run, height, weight, biceps skinfold, triceps skinfold, subscapular skinfold, supra-iliac skinfold, elbow breadth, wrist breadth, calf circumference, mid-thigh circumference, arm length, leg length, shoulder breadth, pelvic breadth, femoral breadth and ankle breadth.

### Measuring procedure

PA was measured using a multi-sensor device SWA (Bodymedia SenseWear Pro Armband; BodyMedia Inc., Pittsburgh, USA). SWA device is based on the recognition of energy expenditure patterns and the estimation of PA. It uses several non-invasive biometrical sensors which measure various physical indicators (e.g. thermal conduction rate, galvanic skin response, skin

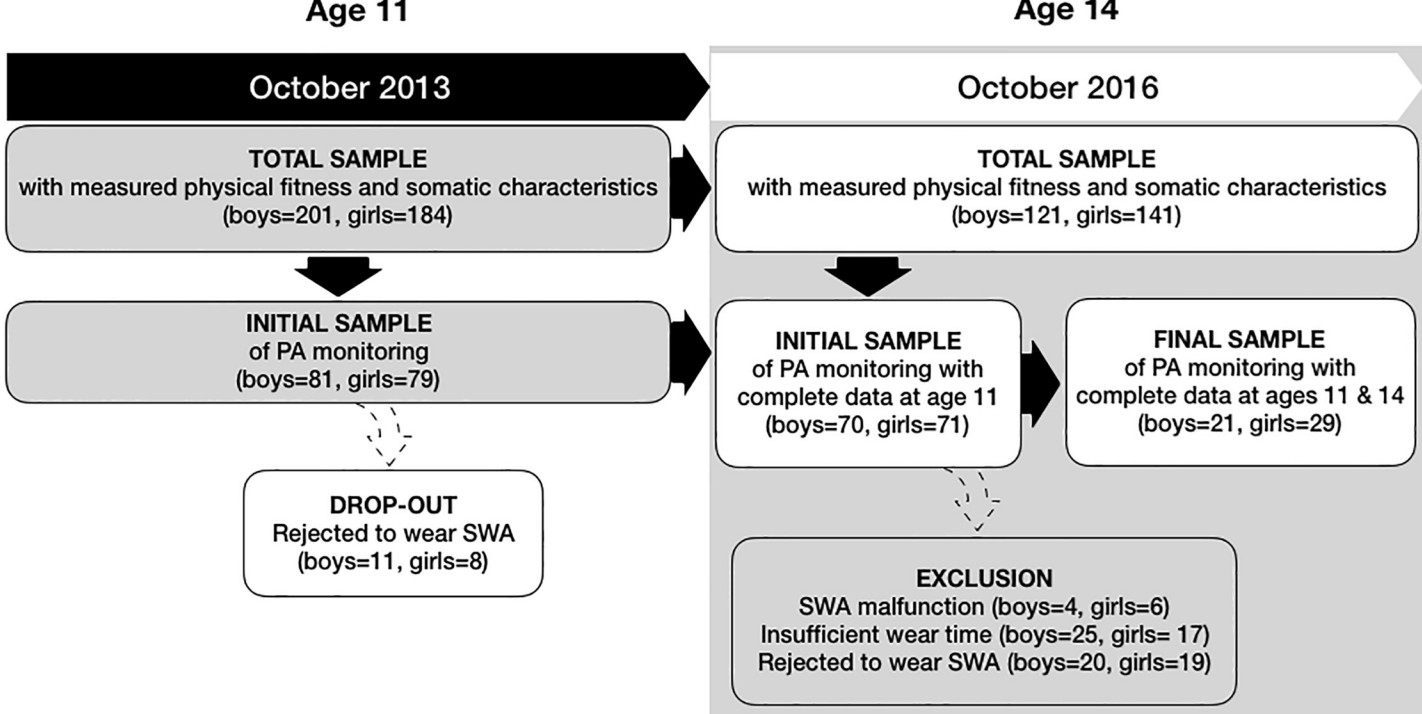

**Fig 1. Participants' flowchart with drop-out and exclusion reasons.**

temperature, environmental temperature, and PA measured with 2-axis accelerometer) and calculates energy expenditure via algorithms, which include data from several sensors as well as sex, age, height and weight of a measured subject. SWA was validated against doubly labeled water in children with high correlations for all comparisons (>0.90) [59] and is considered to be a reliable measuring device in children [40,42,47,60]

Children wore the SWA on the triceps of their right arm for one week [61–63], 24 hours per day except when showering, bathing and when practicing sports activities (usually at competitions or swimming), if they were not allowed to wear it due to safety reasons or device limitations (for example water resistance limitations when swimming). Further analysis of PA included only those children who wore the SWA device at least five days in a row (including both weekend days) [61,64,65] and whose wear time exceeded 90%. The SWA device collected data in one-minute epoch intervals. The main reason for our conservative approach was to reduce the error of under- and overestimation of PA due to missing wear-time in a greater extent than in the existing studies.

Body height, sitting height, and body lengths were measured with the use of anthropometers to one-millimetre precision levels (Siber & Hegner, Zurich, Switzerland). Body mass was measured using a portable electronic scale with 100-gram precision (Tanita BWB-800P, Arlington Heights, IL, USA). The precision of measurement was checked every time the scale was moved.

All other anthropometric measurements, physical fitness measurements, protocols, and equipment have been thoroughly described elsewhere [57].

## Data collection

All the parents were notified about the purpose and procedures of data collection and their written consent was collected prior to measurements. Data collection was performed in

October 2013 and October 2016. A trained team of researchers delivered the SWA devices on a Wednesday morning and collected the device the following Tuesday. The functioning of the SWA was explained to the children prior to the start of the monitoring period. In addition, short written guidelines of the proper use of equipment were prepared and given to the children, their parents, and teachers. Physical fitness tests and anthropometric measurements at ages 11 and 14 were administered by a highly trained team of researchers according to the prescribed protocol [57]. During the measurements, children were barefoot and wore light clothes. Approval of the National Medical Ethics Committee was obtained in June 2013 (ID 138/05/136).

## Variables

The analysis of PA included two indicators of PA estimated through the SWA data: total energy expenditure (in calories) and duration of MVPA (in minutes). Although 3 METs (The Metabolic Equivalent of Task) has been widely used as an intensity threshold to distinguish between sedentary and MVPA, there is considerable evidence that 4 MET is a more suitable cut-off point for classifying MVPA in children and adolescents [66,67]; therefore MVPA was defined as a $PA \geq 4$ MET. For the assessment of the timing of adolescent growth spurt, we used the anthropometric measurements that included body height, body mass, sitting height, and age at the time of measurement. We used Mirwald's equation [68] to calculate the timing of adolescent growth spurt of included children. The timing of the adolescent growth spurt provides a benchmark of the maximum growth during early adolescence and provides a common landmark to reflect the occurrence of other body dimension velocities within and between individuals [69]. Adolescent growth spurt typically occurs approximately two years after the onset of puberty [70]. For boys the predictive equation was: maturity offset (years) $_{boys}$ = -9.236 + (0.0002708 * (leg length * sitting height)) + (−0.001663 * (age * leg length)) + (0.007216 * (age * sitting height)) + (0.02292 * (weight ÷ height * 100)). For girls the predictive equation was: maturity offset (years) $_{girls}$ = -9.376 + (0.0001882 * (leg length * sitting height)) + (0.0022 * (age * leg length)) + (0.005841 * (age * sitting height)) + (- 0.002658 * (age * weight)) + (0.07693 * (weight ÷ height * 100)). The equation estimates the time distance from the adolescent growth spurt at the time of measurement, based on the ratio between the length of the torso and the legs. The growth of leg length precedes the growth of the torso, which changes the body proportions and by subtracting maturity offset from chronological age, age at adolescent growth spurt could be determined. This method was validated in studies of youth PA [71–73] and of young athletes [74–78]. Moreover, this method has been validated in an independent longitudinal samples in boys [79] and girls [80].

## Data analysis

PA patterns were analysed with the Bodymedia SenseWear Professional 8.1 software package while all the statistical analyses were calculated by SPSS 25.0 (IBM Inc., Armonk, USA). Data are presented as mean and standard deviation, t-values, F-ratios and effect size (Cohen's d and $\eta^2$). Independent T-test was used to check for statistical differences in multiple physical fitness and anthropometric variables (standing broad jump, obstacle course backwards, 20-s drumming test, flamingo balance test, sit and reach, shoulder circumduction, hand grip strength, bent arm hang, 20-m shuttle-run, height, weight, biceps skinfold, triceps skinfold, subscapular skinfold, suprailiac skinfold, elbow breadth, wrist breadth, calf circumference, mid-thigh circumference, arm length, leg length, shoulder breadth, pelvic breadth, femoral breadth and ankle breadth): a) between boys and girls from the total sample in 2013 (n = 385), who were excluded from (n = 244) or included in (n = 141) the initial sample at age 11, and b) between

boys and girls from the total sample in 2016, who were excluded from (n = 160) or included in (n = 50) the final sample at age 14 (Fig 1). In this way we were able to determine if the 11- and 14-year-olds who were included in PA analysis in 2013 and 2016, respectively, differed from their excluded peers. Differences in MVPA between boys and girls were then analysed using a paired sampled T-test. Differences in maturity between boys and girls were considered as proposed in other studies [73,81] and were analysed using Independent samples T-test, while normal distribution was checked with Q-Q plots, and the homogeneity of variances with Levene's test. Repeated Measures Analysis of Covariance (RM ANCOVA) with Bonferroni correction was used to check for differences in longitudinal measures (2 time points, 3 years apart) between boys and girls, controlling for maturity with time distance from adolescent growth spurt (used as a covariate). ES was calculated using Cohen's d (when t-test was used) and $\eta^2$ (when RM ANCOVA was used). Alpha was set at 0.05.

## Results

Independent samples T-test revealed that there were no statistical differences in physical fitness or somatic characteristics between boys and girls from the total sample (n = 385), who were included in the initial sample (n = 141) or excluded from it (n = 244) at age 11 (see S1 and S2 Tables).

The only small statistically significant difference at age 11 was observed in sit and reach test among boys, included in the initial sample (17.80 cm) and boys, who were excluded from it (15.82 cm) sample (t(190) = -2.03, p = 0.044). At age 14 no statistical differences in physical fitness or somatic characteristics were observed (see S3 and S4 Tables) between boys and girls from the total sample in 2016, who were included in the final sample (n = 50) or excluded from it (n = 160).

The expressed significant difference in the flexibility of hamstrings muscles and lower back (measured with sit-and-reach test) is probably irrelevant in regard to physical activity and it is unlikely that they influenced the drop-out or irregular wearing of the SWA sensors that lead to exclusion from the analysis of the final sample. Estimated PA values of girls and boys at ages 11 and 14 are shown in Fig 2.

Without controlling for the timing of adolescent growth spurt, total MVPA off boys and girls was significantly higher at age 11 than at age 14 (t(49) = 2.75, p < 0.01, d = 0.39). MVPA in girls at age 11 was statistically higher than three years later (t(28) = 3.18, p < 0.01, d = 0.20), whereas the difference in MVPA between boys at ages 11 and 14 was not statistically significant (t(20) = 0.545, p = 0.592, d = 0.13). MVPA in boys at age 14 was significantly higher than in the girls of the same age (t(32.14) = 3.357, p < 0.05 d = 0.092) by 62.4 min, whereas the differences in MVPA between girls and boys at age 11 was not statistically significant (t(32.14) = 0.935, p = 0.355, d = 0.27).

There was a statistical difference in the timing of adolescent growth spurt between boys and girls. Girls experienced it significantly earlier (12.01 ± 1.0 years old) than boys (13.2 ± 0.75 years old) (t(48) = 4.201, p < 0.01, d = 1.35).

After controlling for the timing of the adolescent growth spurt, results of RM ANCOVA showed that there was a significant difference in average MVPA between boys and girls at age 11 (F(1,47) = 6.294, p < 0.05, $\eta^2$ = 0.12). Total difference of boys' and girls' MVPA at age 11 and 14 (F(1,47) = 5.945, p < 0.05, $\eta^2$ = 0.11) proved to be significant. The amount of MVPA of boys at age 14 did not significantly differ from their MVPA at age 11 (p = 0.60), whereas MVPA of girls at age 14 significantly differed from their MVPA at age 11 (F(1,27) = 9.799, p < 0.01, $\eta^2$ = 0.17).

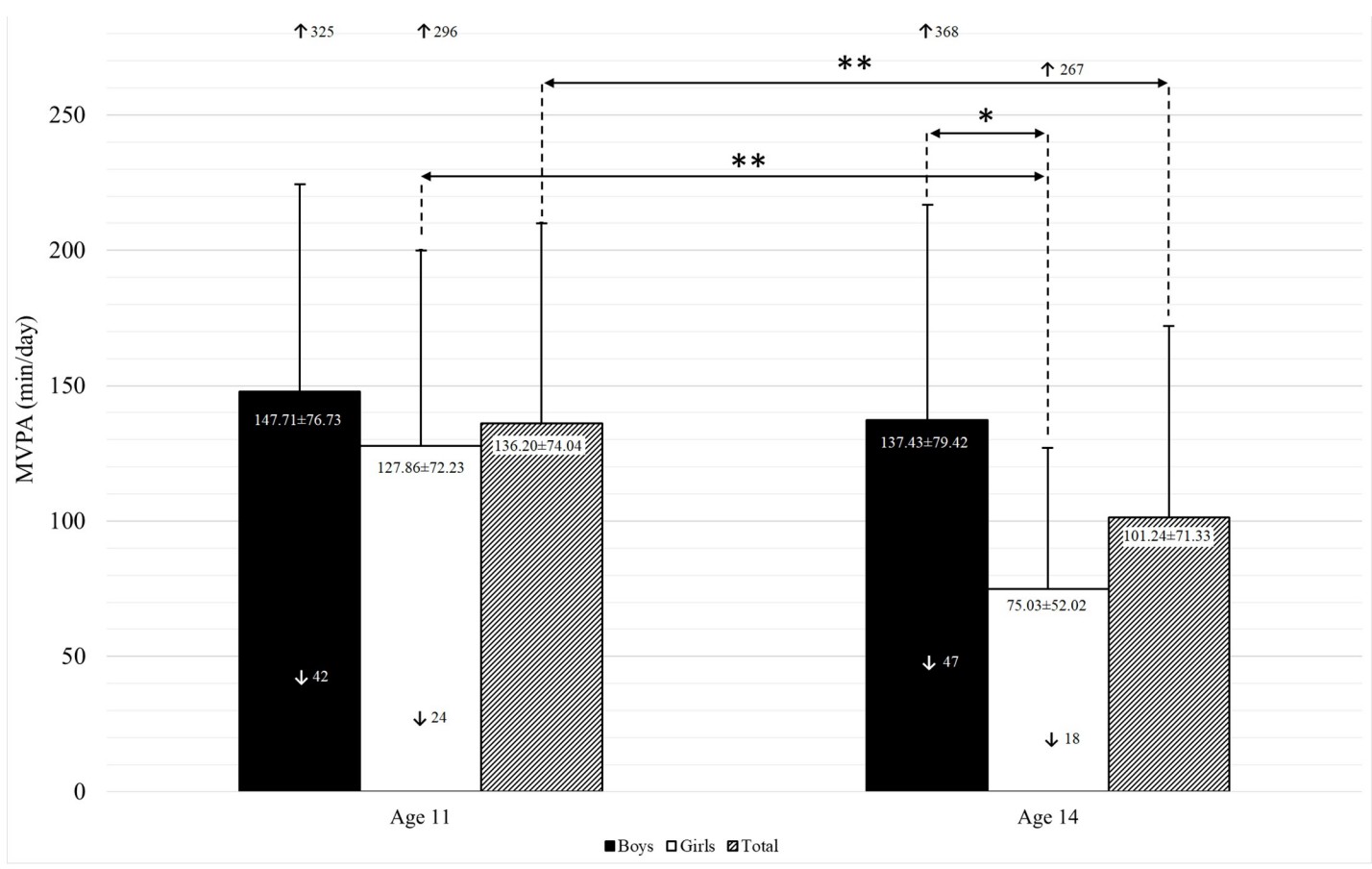

**Fig 2. Estimated MVPA of girls and boys at ages 11 and 14.** MVPA–moderate-to-vigorous physical activity; ↑ maximum; ↓ minimum; * p < 0.05; ** p < 0.01; values in the columns as Mean ± SD.

## Discussion

The aim of this longitudinal study was to analyse the changes in the amount of PA of boys and girls in the early stages of adolescence based on objectively assessed PA data.

The first important finding of the present study is that there is a visible decrease in MVPA between the ages of 11 and 14 which is more pronounced in girls. The average decrease of MVPA in girls was 41% compared to boys, who experienced a 7% decrease. The decrease of MVPA during early adolescence and especially in girls [82,83] was expected since such trends have been observed also in other studies [13,16,84,85]. The findings from the present study concur with a comparative study between Denmark, Portugal, Estonia, and Norway, which discovered decreasing MVPA in pre-adolescents, which was more pronounced in boys (approx. 30%) than in girls (approx. 20%) [86]. Similar findings have also been noticed by Jurak et al. [85] in a comparative study on PA of 11-year-old children from Ljubljana (Slovenia), Zagreb (Croatia) and Ann Arbour (USA). Boys from Ljubljana and Zagreb were physically more active than girls, although more than 90% of adolescents reached the WHO PA recommendations during the weekdays and more than 85% during weekends. Previous studies in Slovenia showed that 86% of the 11-year olds and 66% of 14-year olds is reaching the WHO PA recommendations [49,50,52] which is similar to the findings of the present study, where 84% of 11-year olds (n = 50) and 68% of 14-year olds (n = 50) are reaching the WHO PA recommendations. The significant decrease of MVPA between ages 11 and 14 in Slovenia is likely

a combination of several factors. Although some studies found biological development to be related to the drop of PA in adolescence [56], contributing it to various factors such as non-exercise activity thermogenesis [87,88], reduced dopamine release or loss of dopamine receptors [89–91] or influence of biological maturity status, our analysis showed that the drop was evident with or without controlling for the timing of adolescent growth spurt. This leads to the conclusion that in our case the underlying factors of the adolescent drop of PA might not be biological but environmental. In this regard the school environment can be identified as one of the most important factors for this evident change. Namely, between ages 6 and 11 all Slovenian children have 3 classes of PE per week (45 minutes each), while in the last three years of primary school from ages 12 to 14 the number of compulsory PE classes drops to two per week [92] which is a considerable reduction. In the last three years of primary school pupils are also becoming growingly burdened with schoolwork; the number of school subjects increases from 11 to 14 [85] and their weekly workload increases from 25.5 to 28.5 hours, sometimes even to 32 hours [85]. Consequently, the sedentary activities of children at school and at home increase (more homework and studying), which can partly explain the decrease in PA.

As there are no intergenerational comparisons about the objectively evaluated PA of schoolchildren, it is impossible to conclude whether the 14-year-olds today achieve lower levels of PA than their peers in the past in the past. The comparisons of the level of the PA of Slovenian teenagers with their peers from other countries show that the PA of the Slovenian population is higher, particularly during the week [48–51]. Also the comparisons of physical fitness, which is inevitable result of habitual physical activity, show that Slovenian 14-year-old girls today have significantly higher level of physical fitness than the girls of the same age in 1990, whereas the boys still have a slightly lower physical fitness level [93] than their peers from the same year. All of these findings indicate that the decrease of MVPA in this age period is a universal phenomenon [15,86,94,95], although not necessarily influenced by biological changes during the maturation period.

## Strengths and limitations

In the existing studies, the duration of wear time was ranging from 1h/day to 16.7 h/day [96], while the most common duration period was $\geq$ 10 h/day [97]. In order to assess as complete picture of the pre-adolescents' daily PA patterns as possible one of the most important strengths of the present study is the wear time, which was over 90% (21h and 40min) per day. However, this strong criterion was unfortunately also the main reason for omitting a considerable number of data from the analysis. Further strengths of the present study is the use of the same measuring equipment throughout the study and the inclusion of timing of adolescent growth spurt. However, there are also some limitations: i) as in all PA studies, children were aware that they were being measured and monitored, which might have resulted in the changes of their habitual PA patterns; ii) high measurement and equipment costs limited the availability of SWA sensors and did not allow the measurement of PA of all included children during the same week which means that the PA patterns could be affected by other factors such as different weather conditions; iii) due to the short battery life and recording memory, a 1-minute interval for the collection of signal was used; consequently, not all short-burst PA durations could have been recorded, which might have resulted in underestimating the overall duration and intensity of recorded PA; iv) MVPA was measured only for one week, therefore we cannot interpret our results as behavioural PA; v) biomechanical and physical factors could affect the results as increased mass give lower acceleration and the increased height of children produces longer pendulum [98]; vi) arm worn device is more sensitive to arm movement, therefore results could be overrated [99]; vii) external factors could influence PA between the

ages of 11 and 14: family problems, friendship problems, dietary habits and changes of environment/school, socioeconomic status, which have not been controlled for; viii) rigorous inclusion criteria regarding the wear time could have resulted in exclusion of some children who intensively practice sport; for example, children had to remove the measurement equipment during the competitions, in contact sports, and swimming activities; ix) total body mass was used in calculation of energy expenditure, thus it could be confounded by adiposity (a more precise choice would be lean mass proportional measures but it could not be used because the algorithms of the PA analysis software are based on actual body mass); x) anthropometry and equation-based methods like Mirwald et al. equation [69] for predicting the timing of adolescent growth spurt are not the most precise tools [100], therefore radiograph-based methods [80,101] would be recommended in future research; xi) the use of the fixed intensity thresholds in assessment of MVPA during pre-adolescence may not be the most suitable approach during adolescence due to changes in body size, composition and physical fitness, therefore the use of specific cut-off points should be used in the future [102].

## Conclusion

This study contributes to the understanding of changes in PA among Slovenian pre-adolescents. Similar to other studies our results confirm the decrease of MVPA with age, but the differences between boys and girls in our study are considerably more pronounced. Since the greater drop of MVPA was observed in girls, future research should aim to explore the use of different approaches in encouraging the PA in boys and girls. Future studies should continue prioritizing longitudinal methodologies with bigger implications on gender differences in MVPA, including information about other PA determinants, such as environment, socioeconomic status and parental education.

## Supporting information

**S1 Table. Independent samples T-test comparison of physical fitness and somatic characteristics between boys from the total sample who were excluded (n = 120) and the ones who were included (n = 81) in the initial sample at age 11.**
(DOCX)

**S2 Table. Independent samples T-test comparison of physical fitness and somatic characteristics between girls from the total sample who were excluded (n = 105) and the ones who were included (n = 79) in the initial sample at age 11.**
(DOCX)

**S3 Table. Independent samples T-test comparison of physical fitness and somatic characteristics between boys from the original sample who were excluded (n = 84) and the ones who were included (n = 21) in the final sample at age 14.**
(DOCX)

**S4 Table. Independent samples T-test comparison of physical fitness and somatic characteristics between girls from the original sample who were excluded (n = 102) and the ones who were included (n = 29) in the final sample at age 14.**
(DOCX)

**S1 Data.**
(XLSX)

## Author Contributions

**Conceptualization:** Vedrana Sember, Gregor Starc.

**Data curation:** Vedrana Sember, Gregor Starc.

**Formal analysis:** Vedrana Sember, Gregor Starc.

**Investigation:** Vedrana Sember, Gregor Jurak, Marjeta Kovač, Gregor Starc.

**Methodology:** Gregor Starc.

**Project administration:** Gregor Jurak.

**Supervision:** Gregor Starc.

**Visualization:** Saša Đurić.

**Writing – original draft:** Vedrana Sember, Gregor Starc.

**Writing – review & editing:** Vedrana Sember, Gregor Jurak, Marjeta Kovač, Saša Đurić, Gregor Starc.

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
