## [Decision Letter · Decision Letter 0]

4 Oct 2019

PONE-D-19-20032

Decline of physical activity from childhood to adolescence: A 3-year longitudinal study

PLOS ONE

Dear Prof. Starc,

Thank you for submitting your manuscript to PLOS ONE. After careful consideration, we feel that it has merit but does not fully meet PLOS ONE’s publication criteria as it currently stands. Therefore, we invite you to submit a revised version of the manuscript that addresses the points raised during the review process.

Please see below for extensive and detailed comments from two expert reviewers. While both reviewers see interest and merit in your study, they have comments and concerns about various aspects of the paper, all of which should be addressed fully in your revision.

We would appreciate receiving your revised manuscript by Nov 18 2019 11:59PM. To enhance the reproducibility of your results, we recommend that if applicable you deposit your laboratory protocols in protocols.io, where a protocol can be assigned its own identifier (DOI) such that it can be cited independently in the future. For instructions see: http://journals.plos.org/plosone/s/submission-guidelines#loc-laboratory-protocols

We look forward to receiving your revised manuscript.

Kind regards,

Kathryn L. Weston, PhD

Academic Editor

PLOS ONE

Journal Requirements:

Additional Editor Comments (if provided):

Reviewers' comments:

Reviewer's Responses to Questions

**Comments to the Author**

1. Is the manuscript technically sound, and do the data support the conclusions?

Reviewer #1: Partly

Reviewer #2: Partly

2. Has the statistical analysis been performed appropriately and rigorously? 

Reviewer #1: No

Reviewer #2: No

3. Have the authors made all data underlying the findings in their manuscript fully available?

Reviewer #1: Yes

Reviewer #2: Yes

4. Is the manuscript presented in an intelligible fashion and written in standard English?

Reviewer #1: No

Reviewer #2: Yes

5. Review Comments to the Author

Reviewer #1: Oversell, the results of the study are relevant in determining PA levels of boys and girls in Slovenia. However, the paper requires some major modifications in terms of how the data is reported and interpreted, to ensure clarity to the reader. The discussion and conclusion should support the findings, without overstating or establishing any causal inference.

Abstract

Line 17: please change ‘The study’ to ‘This study’

Line 18: please change to ‘boys and girls aged 11 and 14’. Please state n = for each age. From this sentence, it is not clear if the 11 year olds and 14 year olds were the same children (i.e. you analyzed their PA at aged 11, and then followed up 3 years later by analyzing their PA levels again at aged 14). Please make this clear to the reader. Also, can you please state how long they wore the Bodymedia for (i.e. how many days), and if the measurements were taken at the same time of year (please provide month and date)

Line 20-21: It is not clear about when PA was measured, and what you mean by subsample at the age 11, and final subsample at age 11 and 14. Can you please re-word to clarify this for the reader

Line 21: please use abbreviation (MVPA)

Line 22: please clarify what you mean by ‘longitudinally measured’ and provide n =. What ages are you referring to? Both 11 and 14 years? Or just 11 years? And how did you compare?

Line 23: please add ‘a’ before representative, and delete ‘the’ before age 11

Line 24: please use abbreviation (MVPA)

Line 24 – 24: compared to what?

Lines 24-26: please provide p values. The results section is not clear for the reader. You have ES, the eta squared symbol, and p values. Can you please provide a concise statement about the results, and use consistent reporting values (i.e. p values, ES, t values, f ratios etc)

Line 28: please use abbreviation (MVPA)

Line 28-29: please use abbreviation (MVPA)

Lines 30-32: The conclusions do not match the aim or findings. Can you please reword to include an appropriate conclusion based on the findings, as you do not directly measure schoolwork, participation in PE lessons, or extracurricular PA/PA outside the school environment

Introduction

Line 35: please delete the full stop and ‘it’, and replace with ‘and’ is one of the …..

Line 41: please add ‘the’ before adult

Line 42: please insert a comma before which

Line 43: please write World Health Organization in full before abbreviating

Line 48: ‘suitable contents’ is not clear for the reader. Can you please define

Lines 49 – 52: is there any evidence that the guidelines should differentiate between childhood and adolescence? Could you please clarify what you mean by childhood and adolescence (i.e. provide the ages in years). Is there any evidence that you can provide to support increased sedentary behavior in adolescence?

Lines 52-54: can you please provide the specific ages of childhood and adolescence (see comment above), and also provide evidence to support why it is important to analyse these changes. What does the literature state in terms of the transition from childhood and adolescence? How does this impact PA in adulthood etc.

Lines 54-56: I am not sure that ‘the matter is further complicated’ in relation to your study, as you are not comparing children from different countries, only Slovenia. So I would delete ‘the matter is further complicated’. You also use the word ‘different’ three times in this sentence. Could you please reword to allow the sentence to flow better for the reader

Line 59: please provide evidence that the age groups you are referring to are the ‘most problematic’

Line 63 delete ‘the’ before recent

Paragraph two (lines 60 – 78): I am not sure on the relevance of the paragraph to your study. You discuss advantages and limitations of questionnaires, DLW and accelerometers (none of which are utilized in your study). But you do not mention wearable technologies. And then the paragraph suddenly finishes, and moves on to discuss PA in Slovenia, which does not flow. As you are not comparing these methods (i.e. questionnaires, DLW or accelerometers) in your study I would remove this paragraph, and include pertinent and relevant information on wearable technologies and the use of the Body Media Sense wear armband to assess PA in children

Line 79: please specify the ages of the school children

Lines 81-86: Here you define preadolescents, and puberty. However, in the first paragraph (lines 50-51) you talk about the difference between childhood and adolescence. As you are only looking at PA in children aged 11 and 14, I do not think you can refer to ‘issues/problems’ between childhood and adolescence (in the first paragraph), as you are not actually measuring this in your study. The world health organization defines adolescence as between the ages of 10 and 19. Early adolescence (i.e. puberty) may be defined as ages 10-14. I would suggest therefore, that you temper your introduction (particularly the first paragraph), as you are not really examining differences in PA between childhood and adolescence, but looking at the change in PA during early adolescence (i.e. puberty)

Line 81-82: please see comment above. Are you sure that 15-18 y is ‘preadolescent’? Or is this age group mid adolescence? My understanding is that 10-14 y is ‘preadolescent’.

Lines 83-91: This information would be better suited to the methodology section. It is also not clear from these sentences exactly how you will control for biological maturity. What do you mean by discriminating factor? How will you control for this in the analyses? Please state this clearly in the methodology section only (i.e. PHV) was entered as a covariate

Can you please provide a hypothesis. And also state how long you will measure PA for. You mention ‘longitudinally’. But do you just mean you will analyze their PA once when they are 11, and once again when they are 14. How long will you analyse PA for?

Methods

Line 96. What is ACDSi. Please define this abbreviation

Line 98. What do you mean by ‘multiple location approach’. Was their random stratified sampling from each area of the country?

Line 103. How were the subsample of n = 160 chosen? Was their random stratified sampling of the 385 children eligible?

Line 107: please delete of before 14

Line 108-109: how did you define sufficient wear time?

Line 128: delete ‘also’ before in

Line 129: why was a 7 d period chosen? Due to logistical reasons (i.e. amount of kit)? Does 7 d provide a reliable and valid measure of PA in these children?

Lines 132-134. What is this cut off based on? Why 5 d? and why > 90%?

Line 148: To have worn the devices for 7 d should the collection not have bene the following Wednesday? Or am I missing something

Lines 166 – 172. It is a little difficult to follow as a reader, who did not conduct the study, what you mean by first to second round of PA. Is this from October 2013 (i.e. aged 11), to October 2016 (i.e. aged 14)? What do you mean ‘final subsample’, and ‘initial subsample’, and ‘total sample’. Please be very clear and specific about the time (i.e. month/year), number and ages of children, sex etc that you are referring to

Line 171: please insert ‘the’ before following

Line 181: please state which variables, rather than just ‘multiple variables’

Line 182: provide n for ‘total sample’, provide n = for subsample, provide n = for subsample at age 11. I found it difficult to read this sentence, and to follow exactly which samples/children you are referring to.

Lines 182-187 may be more useful in line 165

Line 188: please state that SPSS was used in the first sentence, prior to stating a t-test was conducted

Can you please state how data is presented (e.g. mean and SD), and how effect sizes were calculated

Results:

Lines 198-200. Again this sentence is difficult to follow as a reader. Are you just repeating what you have mentioned in the methods section? Please be very clear and specific about the time (i.e. month/year), number and ages of children, sex etc that you are referring to. I would delete lines 198-200

Line 199: insert ‘the’ before subsample

Line 201. Please define ‘groups’

Line 205: please subscript the degrees of freedom

Line 208: can you please provide a citation to support his

Table 1: can you please removed the borders in the middle of the table, and only include borders at the top and bottom. Can you please state what you mean by average (was this the mean). Please include the mean and SD in one column only. Please include the minimum and maximum in once column only. Is there a reason why you are reporting both min-max and mean (SD)? Please use the SI units for age (i.e. y)

Line 215: when you state sig more PA. Do you mean MVPA was higher at age 11 compared to age 14? Or are you talking about total EE? Please format the parentheses appropriately and include the ES in the same parenthesis. Is this eta squared ES?

Please include ES for all analyses

Please be consistent with your terminology. You state on line 217 the ‘difference in MVPA’. Whereas again on line 218 you refer to ‘more active’. As you are not measuring overall PA, but looking at MVPA minutes, please use MVPA consistently, and remove any reference to ‘more physical activity’. I cannot see any reference to total EE. Is this what you mean by ‘more physically active’ – please use consistent terminology relevant to your outcome measures

Line 221 – 222.: please delete this sentence. Please make the results concise and clear. You have already described the analyses conducted in the analyses section, so you do not need to repeat again how the analyses was conducted

Line 222: Delete independent sample t test, and just sate what was found

Line 227: p<0.5 would not be significant. Is this a typing error?

Lines 227-229: can you please include the eta square value in the same parenthesis

Line 232: please delete

Discussion

Line 237: please delete the word critical, as we cannot determine causal inference that it is indeed critical. By PA do you mean MVPA or TEE?

Line 239: please delete the space between 7 and %. Can you please discuses the implications of an average of 40 min reduction in MVPA for girls aged 11 – 14, and a 10 min reduction for boys. Please discuss those who had very low MVPA (I.E. 24 or 18 min). What % of 11 and 14 year old boys and girls met the PA recommendations for Slovenia? The SD are rather large – suggesting a great deal of variance in the PA levels. Can you postulate why this is. Do you have any information on which schools these children came from, what areas they live in, their socioeconomic background etc?

Lines 264-269: This sentence is very long. Can you please re-word to make it easier for the reader to follow, as numerous thoughts and points are raised in this sentence.

Line 273: what do you mean by more efficient than girls of the same age in 1990?

Lines 276-277. I am not sure you can make such a confident statement given you did not directly measure any changes in physical fitness

Lines 278-280: Can you please reword this sentence. I do not understand what you mean by ‘the question arises as to how big a drop in PA levels from childhood does influence the physical efficiency of children when they reach adolescence’. What do you mean by efficiency? And children compared to adolescence? You only measured MVPA and total EE in 11 and 14 year olds.

Lines 281-284: as you did not directly measure PA and PE lessons, or support from friends etc, can you please reword/temper these sentences and relate directly to your findings.

Overall, the discussion needs some work in terms of the flow of sentences, and the tempering/relevance of the findings (see comments above on reporting total EE, discussing if differences are meaningful, reporting % of children that met PA guidelines etc). You cannot make any causal inference for some of the points you make (with regard to PE lesson, maturity etc), as you only looked at 7 d of MVPA at age 11 and age 14. Please ensure the discussion only directly relates to your findings. And any postulations are interpreted with caution.

Line 294-295. Please reword. It is not clear that you are discussing the wear time of the SWA in the present study compared to previous research.

Lines 300-301. Can you make this sentence more concise. You mention consistent measurement equipment and criteria separately, objective EE assessment. Are you simply referring to using the SWA with the PHV. Please use specific and consistent terminology, and do not overstate the measurements. Please provide the reference for the SWA being valid and reliable for MVPA and EE use over 7 d in children. With regards to the limitations, please also acknowledge only analyzing the PA for one week. You do not mention any data collection in terms of PE lesson, outside school PA or exercise, illness etc which may have been important for interpreting the findings

Line 321. Please insert pre-adolescents. Please replace similarly with similar

Line 323-324. You cannot determine causal inference from puberty (i..e PHV) and MVPA. It was only inserted as the covariate. It may not have been the reason why girls decreased their MVPA. I would suggest that future research needs to explore (quantitatively and qualitatively) the reasons why girls are decreasing their MVPA in Slovenia.

Lines 326 – 330. Can you please temper and/or remove these sentences. As they are not measured or related to the outcomes in your study. Please ensure the conclusion is relevant to the discussion and results

Reviewer #2: I was asked to read an interesting manuscript titled “Decline of physical activity from childhood to adolescence: A 3-year longitudinal study” by Dr. Sember et al. The study investigated the changes in physical activity during 3y follow-up period in adolescents and showed interestingly that MVPA declined in girls but not in boys.

General comments

Although the introduction describes important areas related to this study, it does not adequately describe the available evidence on the decline of physical activity from previous studies. The Authors also bring some ideas that maturation is important in study of changes in physical activity, but do not explain why and what is the evidence. Please revise the introduction to better cover previous evidence and to explain what is needed and what gaps this study fills.

The Authors used a physical activity monitor that was used continuously trough 24 hours. Why the Authors only study MVPA and not sedentary behaviour and sleep. That would strengthen the paper significantly.

Specific comments

L79-81. The children in Slovenia are one of the most active paediatric populations. Please define how this was measured.

L95-115. A lot of children dropped off the study How representative this sample was regarding available characteristics such as socioeconomic status, body composition and size (e.g. BMI percentile/sd-score, body fat percentage). Please define these information here and not just in the results.

L127-128. Can the Authors define how SWA was validated. Was it validated against doubly labelled water? What was the agreement between methods. Please expand this information.

L134-135. Please comment the epoch length. Can the Authors show that 60 s epoch length can capture physical activity level reliably. Some studies have shown that 60 s epoch is not able to capture true levels of moderate and vigorous physical activity (1).

L158-163. The Authors need to put more emphasis in the validity of their physical activity data. For example, it is important to describe whether adult derived MET definition (3.5 mL/kg/min) was used or was more appropriate definition used. There are strong data suggesting that adult derived METs are not suitable to used in children (2). Furthermore, METs can produce a significant bias in physical activity assessment (3) and fixed MET thresholds for moderate and vigorous physical activity underestimate true prevalence of physical activity in overweigh and low fit individuals (4). Therefore, I suggest that the Authors carefully describe the procedure used in the study, consider their effect on the results, and highlight these in the limitations.

It seems that the Authors have data on 20 metre shuttle run test. Although it is not the most valid method to assess cardiorespiratory fitness especially during puberty, it could be used to assess maximal endurance capacity in adolescents. Can the Authors use those results to individually calibrate physical activity intensity? This would be important next step in physical activity research.

Some of these limitations are partly covered in the limitations section, but these needs to be expanded.

L163-165. Please describe which formula was used to calculate maturity offset? Mirwald or Moore? Please also describe the whole formula and its validity in this age group.

L182-183. Please describe Mirwald method earlier. Please also note that age at peak height velocity is not equal to onset of puberty which is mostly assessed as transition from stage 1 to stage 2 in stages described by Tanner. Please revise the terminology accordingly.

L192-194. How maturity was used in the model as covariate? It was a bit unclear to me was it only baseline maturity or change in maturity. It is important also to control changes in maturity because at any age children and adolescents mature at different rates.

L193-195. It not clear here whether time x sex interaction were studied or how data was in fact analysed. A figure of results could make it clearer and show the trend of physical activity level and interactions in boys and girls.

L216. Please describe how ES was defined.

L126-232. Does this mean that the results did not change when maturity was adjusted for? It is still a bit unclear whether only baseline maturity was used as a covariate. It would be important to know whether larger changes in maturity (i.e. faster rate of maturation) are related to for example larger decline in physical activity especially in girls. Furthermore, be also aware when interpreting the data using METs that changes in body composition during pubertal development are different in males and females with increased body adiposity in females and decreased adiposity in males. These changes may have an influence on MET-based estimates of physical activity as MET itself is adiposity confounded metric.

L237-239. Although merely speculative, is it possible that the decline of physical activity in girls is just because of increasing adiposity as suggested by Kujala et al. (4). Thus, are the same fixed intensity thresholds valid in assessment of physical activity intensity especially during youth when body size and composition as well as physical fitness change remarkably. Please consider these issues at least in limitations.

L248-258. As the Authors state school is probably important factor influencing physical activity and sedentary time in youth. I suggest that the Authors consider also biological explanations explained for example in the article by Eisenmann and Wickel (5).

L262-264. The Authors state that there were no differences in physical activity at the age of 11 and 14 after adjustment for maturity. I was wondering that wasn’t this the case also without the adjustment. Please clarify. Please also consider my other comments on maturity.

L271-274. Please describe what the Authors mean by physical efficiency.

6. PLOS authors have the option to publish the peer review history of their article (what does this mean?). If published, this will include your full peer review and any attached files.

Reviewer #1: No

Reviewer #2: Yes: Eero Haapala

---

## [Author Response · Author response to Decision Letter 0]

16 Nov 2019

REVIEWER 1

We would like to thank you for your thorough reading and very useful suggestions which we tried to incorporate in the revised version to our best abilities. It was especially useful to build on your in-depth understanding of growth and maturation processes and correct our inconsistencies and ambiguities. We are certain that this helped us improve our work considerably and hope that you will find the revised version as a considerable improvement of our original submission.

Abstract

• Line 17: please change ‘The study’ to ‘This study’ 

We changed it to ‘This study’.

• Line 18: please change to ‘boys and girls aged 11 and 14’. Please state n = for each age. From this sentence, it is not clear if the 11 year olds and 14 year olds were the same children (i.e. you analyzed their PA at aged 11, and then followed up 3 years later by analyzing their PA levels again at aged 14). Please make this clear to the reader. Also, can you please state how long they wore the Bodymedia for (i.e. how many days), and if the measurements were taken at the same time of year (please provide month and date)

We revised the text according to your suggestions. We changed to ‘boys and girls aged 11 and 14’ and we stated n = for each age. We also clarified that we measured the cohort of same children. We also made it clear, that children wore the Bodymedia armband for 7 days in October 2013 and October 2016. 

• Line 20-21: It is not clear about when PA was measured, and what you mean by subsample at the age 11, and final subsample at age 11 and 14. Can you please re-word to clarify this for the reader

We revised the sentence according to your suggestions. 

• Line 21: please use abbreviation (MVPA) 

We followed your suggestion to make the abstract more comprehensible, although the instructions on the PlosOne page are noting that “abstracts should not include citations and abbreviations, if possible”.

• Line 22: please clarify what you mean by ‘longitudinally measured’ and provide n =. What ages are you referring to? Both 11 and 14 years? Or just 11 years? And how did you compare?

We revised the sentence according to your suggestions and made clear that the same cohort was measured at ages 11 and 14. Due to space restrictions we could not provide the details on the measured physical fitness and somatic components in the abstract but this is described in details in the text.

• Line 23: please add ‘a’ before representative, and delete ‘the’ before age 11

We revised the sentence according to your suggestions.

• Line 24: please use abbreviation (MVPA) 

We revised the sentence according to your sugestions.

• Line 24 – 24: compared to what?

We revised the sentence and made clear that we compared the average MVPA between ages 11 and 14.

• Lines 24-26: please provide p values. The results section is not clear for the reader. You have ES, the eta squared symbol, and p values. Can you please provide a concise statement about the results, and use consistent reporting values (i.e. p values, ES, t values, f ratios etc.)

We revised the sentence. We agree with your advice to make reporting consistent throughout the text. We explained use of different ES in the methods section. 

• Line 28: please use abbreviation (MVPA)

We revised the sentence according to your suggestion.

• Line 28-29: please use abbreviation (MVPA)

We revised the sentence according to your suggestion.

• Lines 30-32: The conclusions do not match the aim or findings. Can you please reword to include an appropriate conclusion based on the findings, as you do not directly measure schoolwork, participation in PE lessons, or extracurricular PA/PA outside the school environment

We revised the sentence with the focus on the findings. 

Introduction

• Line 35: please delete the full stop and ‘it’, and replace with ‘and’ is one of the 

Corrected. 

• Line 41: please add ‘the’ before adult

Corrected.

• Line 42: please insert a comma before which

Corrected.

• Line 43: please write World Health Organization in full before abbreviating

Corrected.

• Line 48: ‘suitable contents’ is not clear for the reader. Can you please define

We revised the sentence and describe clearly what was included in the guidelines.

• Lines 49 – 52: is there any evidence that the guidelines should differentiate between childhood and adolescence? Could you please clarify what you mean by childhood and adolescence (i.e. provide the ages in years). Is there any evidence that you can provide to support increased sedentary behavior in adolescence?

We are grateful for this comment as it points to an important rationale of this study, which concerns the increase of sedentary time and decline of MVPA in early adolescence. We provided additional references, reporting the increasingly sedentary behavior in early adolescents. We are not aware of any existing evidence of guidelines that are taking into account the biological maturity of children. Guideliness of some countries (Canada, Finland, Australia) are differentiating between different age groups but they do not provide any evidence for the rationale of differentiation. Following your further comments regarding terms childhood and adolescence we defined our subjects as pre- and early-adolescents. We defined pre- and early-adolescents in methods (sample) chapter. We used the same terminology pre-adolescents, early-adolescents and adolescents throughout the whole manuscript. We changed the title of our study to: “Decline of physical activity in early-adolescence: A 3-year cohort study”

• Lines 52-54: can you please provide the specific ages of childhood and adolescence (see comment above), and also provide evidence to support why it is important to analyse these changes. What does the literature state in terms of the transition from childhood and adolescence? How does this impact PA in adulthood etc.

We appreciate your insightful comment regarding terminology. We defined pre- and early-adolescents in the methods (sample) chapter. We revised this part of the text. 

• Lines 54-56: I am not sure that ‘the matter is further complicated’ in relation to your study, as you are not comparing children from different countries, only Slovenia. So I would delete ‘the matter is further complicated’. You also use the word ‘different’ three times in this sentence. Could you please reword to allow the sentence to flow better for the reader

We revised the sentence according to your suggestions. 

• Line 59: please provide evidence that the age groups you are referring to are the ‘most problematic’

We changed the paragraph and did not describe different age group as more or less problematic but instead characterized the period of early-adolescence as the most intensive period in regard of the most intensive period of PA behavioral change. We also provided evidence citing additional relevant sources: 

REFERENCES 

Currie, C., Molcho, M., Boyce, W., Holstein, B., Torsheim, T., & Richter, M. (2008). Researching health inequalities in adolescents: the development of the Health Behaviour in School-Aged Children (HBSC) family affluence scale. Social science & medicine, 66(6), 1429-1436.

Jiménez-Pavón D, Kelly J, Reilly JJ. Associations between objectively measured habitual physical activity and adiposity in children and adolescents: Systematic review. Int J Pediatr Obes. 2010;5: 3–18. 

Straatmann, V. S., Oliveira, A. J., Rostila, M., & Lopes, C. S. (2016). Changes in physical activity and screen time related to psychological well-being in early adolescence: findings from longitudinal study ELANA. BMC public health, 16(1), 977.

Twisk JWR. Physical activity guidelines for children and adolescents. Sport Med. 2001;31: 617–627. 

• Line 63 delete ‘the’ before recent

Corrected.

• Paragraph two (lines 60 – 78): I am not sure on the relevance of the paragraph to your study. You discuss advantages and limitations of questionnaires, DLW and accelerometers (none of which are utilized in your study). But you do not mention wearable technologies. And then the paragraph suddenly finishes, and moves on to discuss PA in Slovenia, which does not flow. As you are not comparing these methods (i.e. questionnaires, DLW or accelerometers) in your study I would remove this paragraph, and include pertinent and relevant information on wearable technologies and the use of the Body Media Sense wear armband to assess PA in children

We agree with your observation and thoroughly revised the paragraph. We also included information on wearable technologies and use of the BodyMedia SenseWear armband to assess PA in children according to your suggestion. 

• Line 79: please specify the ages of the school children

We specified the age of children which is 6 to 17 years of age. 

• Lines 81-86: Here you define preadolescents, and puberty. However, in the first paragraph (lines 50-51) you talk about the difference between childhood and adolescence. As you are only looking at PA in children aged 11 and 14, I do not think you can refer to ‘issues/problems’ between childhood and adolescence (in the first paragraph), as you are not actually measuring this in your study. The world health organization defines adolescence as between the ages of 10 and 19. Early adolescence (i.e. puberty) may be defined as ages 10-14. I would suggest therefore, that you temper your introduction (particularly the first paragraph), as you are not really examining differences in PA between childhood and adolescence, but looking at the change in PA during early adolescence (i.e. puberty)

We defined pre- and early-adolescents in the methods (sample) chapter. We used the same terminology pre-adolescents, early-adolescents and adolescents throughout the whole manuscript. We tempered our introduction according to your suggestions. 

• Line 81-82: please see comment above. Are you sure that 15-18 y is ‘preadolescent’? Or is this age group mid adolescence? My understanding is that 10-14 y is ‘preadolescent’. 

The [15–18] in the square brackets referred to references, not age groups but still had the potential to confuse the readers. The numeration of references is now changed and this ambiguity resolved.

• Lines 83-91: This information would be better suited to the methodology section. It is also not clear from these sentences exactly how you will control for biological maturity. What do you mean by discriminating factor? How will you control for this in the analyses? Please state this clearly in the methodology section only (i.e. PHV) was entered as a covariate

We deleted this part from introduction part and we included the information in methods section under data analysis. We deleted the phrase “discriminating factor” since it was not used appropriately. At the end of the methods part (under data analysis) we described how we controlled for “biological maturity” – it was used as covariate.

• Can you please provide a hypothesis. And also state how long you will measure PA for. You mention ‘longitudinally’. But do you just mean you will analyze their PA once when they are 11, and once again when they are 14. How long will you analyse PA for?

We provided a hypothesis. We described the measurements and statistical analyses in more details in the methods section. 

Methods

• Line 96. What is ACDSi. Please define this abbreviation

We defined the abbreviation and revised the sentence.

• Line 98. What do you mean by ‘multiple location approach’. Was their random stratified sampling from each area of the country? 

We changed “multiple location approach” to “sentinel approach” which is more appropriate to use in the context of our study. We provided more detailed description of the sampling. 

• Line 103. How were the subsample of n = 160 chosen? Was their random stratified sampling of the 385 children eligible? 

Of the whole sample of 11-year-olds who were invited to wear SWA, only n=160 children returned the informed written consent from parents to wear the SWA for the whole week. We were not able to use random stratified sampling without the risk of seriously reducing the sample size at the time of measurements.

• Line 107: please delete of before 14

Deleted. 

• Line 108-109: how did you define sufficient wear time?

Sufficient wear time is defined in methods section. Following recommendations for measuring PA measuring with multisensory devices and accelerometers are ranging from 3 to 10 days, where 7 days might be reasonable standard for all ages (Cain et al., 2013). Definition of valid day is typically defined by a minimum number of wearing hours but can be determined in other ways such as calculating study-specific ratios (e.g. 70/80 rule, where valid day is 80% of a time period defined by 70% of the sample having data) (Cattelier et al., 2005). From the sample of 160 schoolchildren included for PA measurements, 121 children wore accelerometer for 80% of the day. Three years after, from 160 children 70% of children wore monitor for more than 80% of time. We decided to additionally tighten our inclusion criteria and included only children who wore the monitor for at least 90% of the day at age 11 and three years after. This is why we only included 50 children, who wore the monitor for 90% of time. This is how we ensured real longitudinal design with sufficient wear time and reduced the under- or overestimation of PA due to non-wear time. 

REFERENCES

Cain, K. L., Sallis, J. F., Conway, T. L., Van Dyck, D., & Calhoon, L. (2013). Using accelerometers in youth physical activity studies: a review of methods. Journal of Physical Activity and Health, 10(3), 437-450.

Catellier, d. j., Hannan, p. j., Murray, d. m., Addy, c. l., Conway, t. l., Yang, s., & Rice, j. C. (2005). Imputation of missing data when measuring physical activity by accelerometry. Medicine and science in sports and exercise, 37(11 Suppl), S555.

Sallis, J. F., Buono, M. J., Roby, J. J., Micale, F. G., & Nelson, J. A. (1993). Seven-day recall and other physical activity self-reports in children and adolescents. Medicine and science in sports and exercise, 25(1), 99-108.

Trost, S. G., Pate, R. R., Freedson, P. S., Sallis, J. F., & Taylor, W. C. (2000). Using objective physical activity measures with youth: how many days of monitoring are needed?. Medicine & Science in Sports & Exercise, 32(2), 426.

• Line 128: delete ‘also’ before in

Deleted. 

• Line 129: why was a 7 d period chosen? Due to logistical reasons (i.e. amount of kit)? Does 7 d provide a reliable and valid measure of PA in these children? 

Sufficient wear time is defined in methods section. The recommendations for PA measuring with multisensory devices and accelerometers are ranging from 3 to 10 days, where 5 to 7 days are considered a reasonable standard for all ages (Cain et al., 2013; Sallis et al., 1993; Trost et al., 2000). Seven days provide reliable and valid measure of PA in these children. 

REFERENCES 

Cain, K. L., Sallis, J. F., Conway, T. L., Van Dyck, D., & Calhoon, L. (2013). Using accelerometers in youth physical activity studies: a review of methods. Journal of Physical Activity and Health, 10(3), 437-450.

Sallis, J. F., Buono, M. J., Roby, J. J., Micale, F. G., & Nelson, J. A. (1993). Seven-day recall and other physical activity self-reports in children and adolescents. Medicine and science in sports and exercise, 25(1), 99-108.

Trost, S. G., Pate, R. R., Freedson, P. S., Sallis, J. F., & Taylor, W. C. (2000). Using objective physical activity measures with youth: how many days of monitoring are needed?. Medicine & Science in Sports & Exercise, 32(2), 426.

• Lines 132-134. What is this cut off based on? Why 5 d? and why > 90%?

We based our approach on the consensus for measuring PA among the researchers in the field (Esliger et al., 2005). The recommendations of measuring are ranging from 3 to 7 days, and for school-aged children at least one weekend day is recomended (Esliger et al., 2005; Rowlands et al., 2007). The recommendations state that the analysis should include at least 5 consecutive days. We decided for more rigorous inclusion criteria regarding wear-time (5d and >90%), because we wanted to get a more realistic picture of the whole day physical activity. 

REFERENCES 

Esliger, D. W., Copeland, J. L., Barnes, J. D., & Tremblay, M. S. (2005). Standardizing and optimizing the use of accelerometer data for free-living physical activity monitoring. Journal of Physical Activity and Health, 2(3), 366-383.

Rowlands, A. V., & Eston, R. G. (2007). The measurement and interpretation of children’s physical activity. Journal of sports science & medicine, 6(3), 270.

• Line 148: To have worn the devices for 7 d should the collection not have bene the following Wednesday? Or am I missing something

Children wore the devices from Wednesday morning untill Tuesday evening (6 days through daytime and 6 nights). From Tuesday to Wednesday we needed to change the batteries, collect the data, clean the monitors and prepare them for the children from another school for the following day (Wednesday). Since we looked only for full 5 consecutive days and did not analyse sleep patterns, the methodology did not affect the final results. 

• Lines 166 – 172. It is a little difficult to follow as a reader, who did not conduct the study, what you mean by first to second round of PA. Is this from October 2013 (i.e. aged 11), to October 2016 (i.e. aged 14)? What do you mean ‘final subsample’, and ‘initial subsample’, and ‘total sample’. Please be very clear and specific about the time (i.e. month/year), number and ages of children, sex etc. that you are referring to

Thank you for your remark. We directed readers to our flowchart (Fig. 1) for easier understanding and described the flow of the analysis in more detail.

• Line 171: please insert ‘the’ before following

Inserted. 

• Line 181: please state which variables, rather than just ‘multiple variables’

We listed all the variables. 

• Line 182: provide n for ‘total sample’, provide n = for subsample, provide n = for subsample at age 11. I found it difficult to read this sentence, and to follow exactly which samples/children you are referring to. 

We revised this part and also redirected the readers to Fig.1 (flowchart). 

• Lines 182-187 may be more useful in line 165

We followed your suggestion and moved this part in upper section of the manuscript. 

• Line 188: please state that SPSS was used in the first sentence, prior to stating a t-test was conducted

We rephrased and moved this sentence at the beginning of the paragraph. 

• Can you please state how data is presented (e.g. mean and SD), and how effect sizes were calculated

We added a statement on the presentation of the data and calculation of ES according to your suggestion. 

Results

• Lines 198-200. Again this sentence is difficult to follow as a reader. Are you just repeating what you have mentioned in the methods section? Please be very clear and specific about the time (i.e. month/year), number and ages of children, sex etc that you are referring to. I would delete lines 198-200

We revised the text according to your suggestions.

• Line 199: insert ‘the’ before subsample

We revised this part of the text. 

• Line 201. Please define ‘groups’

We defined the groups and redirected reader to flowchart (Fig.1): 

• Line 205: please subscript the degrees of freedom

Since the journal does not have specific rules on how to report the degrees of freedom we used the standard reporting (t(df)=x.xx, p=x.xxx).

• Line 208: can you please provide a citation to support his

We have revised the sentence since we also changed the analysis at age 11 which showed no statistical differences in girls who were included/excluded from the analysis, onlyu one difference in boys. We were, however, unable to find any evidence for flexibility to be linked to the level of PA. On the contrary, the existing evidence shows that obese children who are less physically active often have greater flexibility due to their low muscular tonus. We, thus could not provide any citation in this case but decided not to discuss the flexibility-PA relation since it is not the focus of the paper.

• Table 1: can you please removed the borders in the middle of the table, and only include borders at the top and bottom. Can you please state what you mean by average (was this the mean). Please include the mean and SD in one column only. Please include the minimum and maximum in once column only. Is there a reason why you are reporting both min-max and mean (SD)? Please use the SI units for age (i.e. y)

We designed our table (which is now S5 Table) according to the PlosOne instructions. By “average” we meant mean. We changed it to mean in the manuscript. We combined mean and standard deviation in one column and minimum and maximum in a separate one. We reported min and max, to give more precise information to reader about children’s MVPA duration. We also deleted a part of the table with energy expenditure, because all further analysis was focused only on MVPA (min). We used SI units. Instead of Table 1 is now in manuscript Fig 2.

• Line 215: when you state sig more PA. Do you mean MVPA was higher at age 11 compared to age 14? Or are you talking about total EE? Please format the parentheses appropriately and include the ES in the same parenthesis. Is this eta squared ES? 

We removed a part of the table with energy expenditure results (now S5 Table). We meant MVPA was higher at age 11 compared to age 14; therefore we reworded the sentence (below). We used Cohen’s d and reported results accordingly.

• Please include ES for all analyses

We included ES for all the analyses according to your suggestion. 

• Please be consistent with your terminology. You state on line 217 the ‘difference in MVPA’. Whereas again on line 218 you refer to ‘more active’. As you are not measuring overall PA, but looking at MVPA minutes, please use MVPA consistently, and remove any reference to ‘more physical activity’. I cannot see any reference to total EE. Is this what you mean by ‘more physically active’ – please use consistent terminology relevant to your outcome measures 

We revised all sentences according to your observation. Since we analyze only MVPA we included only MVPA in Table 1 (now S5 Table) and reported results for MVPA accordingly. 

• Line 221 – 222.: please delete this sentence. Please make the results concise and clear. You have already described the analyses conducted in the analyses section, so you do not need to repeat again how the analyses was conducted 

We deleted tis part. 

• Line 222: Delete independent sample t test, and just sate what was found 

We deleted the first part of the sentence and reported only the results. 

• Line 227: p<0.5 would not be significant. Is this a typing error?

Thank you for noticing this typing error and we changed it to “p<0.05”.

• Lines 227-229: can you please include the eta square value in the same parenthesis 

We included eta square value in the same parenthesis. 

• Line 232: please delete

We agree with the comment and we deleted the line accordingly.

Discussion

• Line 237: please delete the word critical, as we cannot determine causal inference that it is indeed critical. By PA do you mean MVPA or TEE?

We replaced word “critical” with “visible”. When talking about PA and results of our study, we changed everything to “MVPA”. 

By PA we meant MVPA. 

• Line 239: please delete the space between 7 and %. 

We deleted the space. 

• Can you please discuses the implications of an average of 40 min reduction in MVPA for girls aged 11 – 14, and a 10 min reduction for boys. 

We did not control MVPA with any environmental variables and other PA determinants, therefore we cannot be confident about the cause-effect relationship. Future studies should continue prioritizing longitudinal methodologies with bigger implications on gender differences in MVPA, including information about other PA determinants (environment, socioeconomic status, parental education…) (mentioned in conclusion part). 

• Please discuss those who had very low MVPA (I.E. 24 or 18 min). What % of 11 and 14 year old boys and girls met the PA recommendations for Slovenia? 

The very low data belong to a girl who is obese. We rearranged this part of discussion and we provided data for children meeting recommendations from longitudinal sample and separately for children at age 11 and 14. 

• The SD are rather large – suggesting a great deal of variance in the PA levels. Can you postulate why this is. Do you have any information on which schools these children came from, what areas they live in, their socioeconomic background etc?

We agree with the observation and we are aware of the large variances in the PA levels, but the results are in line with other results in Slovenia (Jurak et al., 2015; Kovač, Strel, Jurak, Leskošek, & Dremelj, 2013). Slovenia has a public school system which is very homogeneous in terms of teachers, curricula, infrastructure, quality of PE, etc. and the school environment does not present a discriminating factor. We have limited availability of SES data but our other analyses of this data showed no effect of SES on physical fitness. We are currently working on the geographical data (green spaces, playgrounds, roads…) but since these analyses are extremely complex we do not expect to have any useful data for the next two years.

REFERENCES 

Jurak, G., Sorič, M., Starc, G., Kovač, M., MiŠigoj-Duraković, M., Borer, K., & Strel, J. (2015). School day and weekend patterns of physical activity in urban 11-year-olds: A cross-cultural comparison. American Journal of Human Biology, 27(2), 192–200. https://doi.org/10.1002/ajhb.22637

Kovač, M., Strel, J., Jurak, G., Leskošek, B., & Dremelj, S. (2013). Physical Activity, Physical Fitness Levels, Daily Energy Intake and Some Eating Habits of 11-Year-Old Children. , (1), 127–139.

• Lines 264-269: This sentence is very long. Can you please re-word to make it easier for the reader to follow, as numerous thoughts and points are raised in this sentence.

We revised this part.

• Line 273: what do you mean by more efficient than girls of the same age in 1990?

We substituted “physical efficiency” with “physical fitness” and changed the sentences accordingly. We use the term physically efficient for children who show high levels of physical fitness.

• Lines 276-277. I am not sure you can make such a confident statement given you did not directly measure any changes in physical fitness 

We now included also supplementary tables S3 and S3 which include the results of physical fitness tests but we omitted this statement from the text.

• Lines 278-280: Can you please reword this sentence. I do not understand what you mean by ‘the question arises as to how big a drop in PA levels from childhood does influence the physical efficiency of children when they reach adolescence’. What do you mean by efficiency? And children compared to adolescence? You only measured MVPA and total EE in 11 and 14 year olds. 

We changed “physical efficiency” to “physical fitness” throughout the whole manuscript. We added references for decline of PA in these ages and we deleted the last sentence/question, since we do not want to open a new, extensive chapter of physical fitness and since it is not the aim of present study. We explained the use of “children and adolescents” above. 

• Lines 281-284: as you did not directly measure PA and PE lessons, or support from friends etc, can you please reword/temper these sentences and relate directly to your findings. Overall, the discussion needs some work in terms of the flow of sentences, and the tempering/relevance of the findings (see comments above on reporting total EE, discussing if differences are meaningful, reporting % of children that met PA guidelines etc). You cannot make any causal inference for some of the points you make (with regard to PE lesson, maturity etc.), as you only looked at 7 d of MVPA at age 11 and age 14. Please ensure the discussion only directly relates to your findings. And any postulations are interpreted with caution. 

We omitted the unrelated points from the text. 

• Line 294-295. Please reword. It is not clear that you are discussing the wear time of the SWA in the present study compared to previous research. 

We revised the sentence. 

• Lines 300-301. Can you make this sentence more concise. You mention consistent measurement equipment and criteria separately, objective EE assessment. Are you simply referring to using the SWA with the PHV. Please use specific and consistent terminology, and do not overstate the measurements. 

We revised the text according to your suggestions. 

• Please provide the reference for the SWA being valid and reliable for MVPA and EE use over 7 d in children. With regards to the limitations, please also acknowledge only analyzing the PA for one week. You do not mention any data collection in terms of PE lesson, outside school PA or exercise, illness etc which may have been important for interpreting the findings

We provided the references that proved SWA to be valid and reliable for assessment of MVPA in children and adolescents in introduction and methods part. We also acknowledged in the limitation of measuring PA only for one week. As stated before, Slovenia has a very uniform public educational system and curriculum, and any analyses of PE lessons would not contribute any additional value to the analyses. This paper was focused on the trends of decline and possible effects of maturation and did not control for other cofounders that might influence the results. This is why we did not mention other data collection. 

• Line 321. Please insert pre-adolescents. Please replace similarly with similar

We revised it accordingly to your comment. 

• Line 323-324. You cannot determine causal inference from puberty (i..e PHV) and MVPA. It was only inserted as the covariate. It may not have been the reason why girls decreased their MVPA. I would suggest that future research needs to explore (quantitatively and qualitatively) the reasons why girls are decreasing their MVPA in Slovenia. 

We entirely agree with your observation and have used it to emphasise the inability to ascertain maturation to be underlying reason for decline of PA. 

• Lines 326 – 330. Can you please temper and/or remove these sentences. As they are not measured or related to the outcomes in your study. Please ensure the conclusion is relevant to the discussion and results 

We deleted the mentioned statements according to your suggestion. 

REVIEWER 2

We are very thankful for your very constructive and challenging comments, and especially for the excellent ideas regarding the adjustment of MET thresholds and controlling for body composition. It would be a real pleasure to working with you on our future research and publications because we rarely come across reviewers with such a constructive approach. We tried to address your suggestions to our (and our data) best ability. We believe your thorough reading, in-depth understanding and excellent suggestions helped us to improve the quality of this manuscript. 

General comments 

Although the introduction describes important areas related to this study, it does not adequately describe the available evidence on the decline of physical activity from previous studies. The Authors also bring some ideas that maturation is important in study of changes in physical activity, but do not explain why and what is the evidence. Please revise the introduction to better cover previous evidence and to explain what is needed and what gaps this study fills. 

Thank you for this insight. We additionally described the available evidence on the decline of PA with longitudinal methodologies from previous studies in the introduction part. However, none of the identified longitudinal studies on PA considered the maturation aspect of participants. 

The Authors used a physical activity monitor that was used continuously trough 24 hours. Why the Authors only study MVPA and not sedentary behaviour and sleep. That would strengthen the paper significantly. 

We agree with your suggestion but we were reserved towards the analysis of sleep since SWA monitors have not proven to be the most reliable for sleep detection. We are currently working on another analysis of sedentariness in relation to school-related work and the use of screen technologies and decided not to replicate the same data in two different papers.

Specific comments 

• L79-81. The children in Slovenia are one of the most active paediatric populations. Please define how this was measured.

We described that this was measured objectively and subjectively and we added additional references. 

• L95-115. A lot of children dropped off the study How representative this sample was regarding available characteristics such as socioeconomic status, body composition and size (e.g. BMI percentile/sd-score, body fat percentage). Please define these information here and not just in the results.

Children did not actually droup out of the study, we mostly excluded their data from the analysis because of SWA malfunctions, incomplete data and rigorous demands for wear time (90% of the whole day). Nevertheless we followed the 70/80 rule for wear-time. From 160 schoolchildren, 121 children had complete data for inclusion in further analysis. Three years later some children changed school and did not want to wear SWA anymore. We described in more detail how the sample was constructed from the environmental point of view. In our previous researches in Slovenia SES never proved to be significantly related to either physical fitness or PA of children. We tried to make the physical characteristics of the sample as transparent as possible and we included additional supplemental tables S3 and S4 to show that the sample bias was negligible. 

• L127-128. Can the Authors define how SWA was validated. Was it validated against doubly labelled water? What was the agreement between methods. Please expand this information.

SWA was validated against doubly labelled water in children with high correlations for all comparisons (>0.90). We expanded this information in manuscript and we also added additional references. 

• L134-135. Please comment the epoch length. Can the Authors show that 60 s epoch length can capture physical activity level reliably. Some studies have shown that 60 s epoch is not able to capture true levels of moderate and vigorous physical activity (1).

We are aware of the importance of shorter epoch length, therefore we added one-minute epoch length to the limitations of our study. Nevertheless, we measured MVPA with same equipment and epoch length in both time points. Review articles are indicating that 60 second epoch length is still one of the most common used in youth PA studies, which is good in terms of comparisons. The actual reasons for not using shorter epoch are deriving from the SWA monitor itself. Namely, the manufacturer itself stated that the one-minute epoch algorithm is reliable but that using shorter epochs is not reliable. The second reason is that the use of shorter epochs would decrease battery life and use more memory which means that we would not be able to use the monitors for six consecutive days and this presented serious logisitc problem. 

REFERENCE 

Cain, K. L., Sallis, J. F., Conway, T. L., Van Dyck, D., & Calhoon, L. (2013). Using accelerometers in youth physical activity studies: a review of methods. Journal of Physical Activity and Health, 10(3), 437-450.

• L158-163. The Authors need to put more emphasis in the validity of their physical activity data. For example, it is important to describe whether adult derived MET definition (3.5 mL/kg/min) was used or was more appropriate definition used. There are strong data suggesting that adult derived METs are not suitable to used in children (2). 

We are very thankful for raising these concerns. In the cited article (3), there is a statement that lean mass proportional measures of energy expenditure would enable a more truthful choice to assess physical activity. We used 4 MET as a threshold of MVPA which is a stricter criterion than the usual 3 MET threshold and we think we have been very conservative in this regard. 

• Furthermore, METs can produce a significant bias in physical activity assessment (3) and fixed MET thresholds for moderate and vigorous physical activity underestimate true prevalence of physical activity in overweigh and low fit individuals (4). Therefore, I suggest that the Authors carefully describe the procedure used in the study, consider their effect on the results, and highlight these in the limitations.

We acknowledge that we (i.e. SWA software algorithm) used body mass (and not lean mass), hence we included this into limitations of the study. It is not possible to extract raw data to recalculate energy expenditure and control data for obesity or fitness level from the SWA data. Regarding the fixed MET thresholds, the SWA software uses variables such as sex, age, height and weight of a measured subject and has algorithms, adjusted to children. 

• It seems that the Authors have data on 20 metre shuttle run test. Although it is not the most valid method to assess cardiorespiratory fitness especially during puberty, it could be used to assess maximal endurance capacity in adolescents. Can the Authors use those results to individually calibrate physical activity intensity? This would be important next step in physical activity research. Some of these limitations are partly covered in the limitations section, but these needs to be expanded.

This is a superb and inovative idea, and we would very much like to use this approach in the future, but we think that such an analysis deserves a separate and focused analysis. 

• L163-165. Please describe which formula was used to calculate maturity offset? Mirwald or Moore? Please also describe the whole formula and its validity in this age group.

We revised this part of methods according ti your suggestions. We used and described the Mirwald's equation for maturity offset calculations and we also wrote both formulas – for boys and girls. Application of the used equations was recommended for boys between 10 – 18 years (Mirwald et al., 2012; see also Sherar, Mirwald, Baxter-Jones & Thomis, 2005). The same formulas have been used in studies of youth physical activity (Beets, Vogel, Forlaw, Pitetti, & Cardinal, 2006; Nurmi-Lawton et al., 2004; Weeks & Beck, 2012; Wickel, Eisenmann, & Welk, 2009) and of young athletes (Malina et al., 2012; Matthys, Vaeyens, Coelho e Silva, Lenoir, & Philippaerts, 2012; Sherar, Baxter-Jones, Faulkner, & Russell, 2007; Till, Cobley, O‘Hara, Chapman, & Cooke, 2010; Vandendriessche et al., 2012). Moreover, this method has been validated in an independent longitudinal samples in girls (Malina et al., 2006) and in boys Malina and Kozieł, 2014). 

REFERENCES: 

Beets, M. W., Vogel, R., Forlaw, L., Pitetti, K. H., & Cardinal, B. J. (2006). Social support and youth physical activity: the role of provider and type. American Journal of Health Behavior, 30(3), 278-289.

Mirwald, R. L., Baxter-Jones, A. D., Bailey, D. A., & Beunen, G. P. (2002). An assessment of maturity from anthropometric measurements. Medicine & science in sports & exercise, 34(4), 689-694.

Matthys, S. P. J., Vaeyens, R., Coelho-e-Silva, M. J., Lenoir, M., & Philippaerts, R. (2012). The contribution of growth and maturation in the functional capacity and skill performance of male adolescent handball players. International journal of sports medicine, 33(07), 543-549.

Malina, R. M., Claessens, A. L., Van, K. A., Thomis, M., Lefevre, J., Philippaerts, R., & Beunen, G. P. (2006). Maturity offset in gymnasts: application of a prediction equation. Medicine and science in sports and exercise, 38(7), 1342-1347.

Nurmi‐Lawton, J. A., Baxter‐Jones, A. D., Mirwald, R. L., Bishop, J. A., Taylor, P., Cooper, C., & New, S. A. (2004). Evidence of sustained skeletal benefits from impact‐loading exercise in young females: a 3‐year longitudinal study. Journal of Bone and Mineral Research, 19(2), 314-322.

Sherar, L. B., Mirwald, R. L., Baxter-Jones, A. D., & Thomis, M. (2005). Prediction of adult height using maturity-based cumulative height velocity curves. The Journal of pediatrics, 147(4), 508-514.

Sherar, L. B., Baxter-Jones, A. D., Faulkner, R. A., & Russell, K. W. (2007). Do physical maturity and birth date predict talent in male youth ice hockey players?. Journal of sports sciences, 25(8), 879-886.

Wickel, E. E., Eisenmann, J. C., & Welk, G. J. (2009). Maturity-related variation in moderate-to-vigorous physical activity among 9–14 year olds. Journal of Physical Activity and Health, 6(5), 597-605.

Till, K., Cobley, S., Wattie, N., O'Hara, J., Cooke, C., & Chapman, C. (2010). The prevalence, influential factors and mechanisms of relative age effects in UK Rugby League. Scandinavian journal of medicine & science in sports, 20(2), 320-329.

Vandendriessche, J. B., Vaeyens, R., Vandorpe, B., Lenoir, M., Lefevre, J., & Philippaerts, R. M. (2012). Biological maturation, morphology, fitness, and motor coordination as part of a selection strategy in the search for international youth soccer players (age 15–16 years). Journal of Sports Sciences, 30(15), 1695-1703.

• L182-183. Please describe Mirwald method earlier. Please also note that age at peak height velocity is not equal to onset of puberty which is mostly assessed as transition from stage 1 to stage 2 in stages described by Tanner. Please revise the terminology accordingly.

We described the Mirwald equation earlier in the chapter Methods (variables). We also revised the terminology according to your suggestions and used it consistently. We decided to use adolescent growth spurt instead of peak height velocity and acknowledged that adolescent growth spurt typically occurs approximately 2 years after the onset of puberty.

• L192-194. How maturity was used in the model as covariate? It was a bit unclear to me was it only baseline maturity or change in maturity. It is important also to control changes in maturity because at any age children and adolescents mature at different rates.

We used only baseline maturity. We were not able to control for the tempo of maturity since the use of Tanner's methods for assesment of maturity would result in a serious drop-out and jeopardised our entire study, which was an unacceptable risk since the ACDSi study is a study with the longest tradition in Slovenia. It is becoming extremely difficult to use Tanner's methods like we used to thirty years ago since children and also parents today find it as a serious breach of personal integrity. We tried to use the next best non-invasive measure and this meant we had to rely on an indicator, based on the leg length to body height ratio. 

• L193-195. It not clear here whether time x sex interaction were studied or how data was in fact analysed. A figure of results could make it clearer and show the trend of physical activity level and interactions in boys and girls.

We agree with your observation therefore we present Fig 2 instead of Table 1 (now S5 Table) to graphicaly present the interaction.

• L216. Please describe how ES was defined.

We added a statement on how our data is presented (below) and at the end of the Data analysis section, we described how ES was calculated. We also calculated ES for all the analyses.

• L126-232. Does this mean that the results did not change when maturity was adjusted for? It is still a bit unclear whether only baseline maturity was used as a covariate. It would be important to know whether larger changes in maturity (i.e. faster rate of maturation) are related to for example larger decline in physical activity especially in girls. Furthermore, be also aware when interpreting the data using METs that changes in body composition during pubertal development are different in males and females with increased body adiposity in females and decreased adiposity in males. These changes may have an influence on MET-based estimates of physical activity as MET itself is adiposity confounded metric.

You are right on the target. The results, in fact did not change when maturity was adjusted for (we used baseline maturity as a covariate). This seems to indicate that changes in maturity might not be the underlying cause of PA decline in adolescence and that we should probably focus our attention more on environmantal factors. If this is in fact true, this would mean that we could develop and implement the adequate changes in the adolescent environment to prevent the decline. If the mechanism was underlied by biological development, thare would not be much we could do to tackle the problem. You expressed another terrific idea of adjusting energy expenditure with body composition, which is a problem that deserves a separate analysis and is currently ahead of time. In our view it would be extremely important ti exclude fat mass from the energy expenditure estimations in the future since energy is spent only in muscle cells. Fat mass is dead mass in this regard. Since METs that were calculated by SWA software were estimated from, gender, age weight and height, we wouldn't classify MET as an adiposity confounded metric since it does not differentiate between fat and lean mass. 

• L237-239. Although merely speculative, is it possible that the decline of physical activity in girls is just because of increasing adiposity as suggested by Kujala et al. (4). Thus, are the same fixed intensity thresholds valid in assessment of physical activity intensity especially during youth when body size and composition as well as physical fitness change remarkably. Please consider these issues at least in limitations.

This is an interesting idea but we are not confident what is cause and effect here. It would be more logical to consider growing adiposity as an indicator of lower habitual PA. It makes perfect sense that different people experience the same intensity of PA differently and that the same intensity does not mean the same level of PA for different people. We, nevertheless, included your suggestions in the limitation part. 

• L248-258. As the Authors state school is probably important factor influencing physical activity and sedentary time in youth. I suggest that the Authors consider also biological explanations explained for example in the article by Eisenmann and Wickel (5).

We included your suggestion for other biological factor affecting PA in the discussion part.

• L262-264. The Authors state that there were no differences in physical activity at the age of 11 and 14 after adjustment for maturity. I was wondering that wasn’t this the case also without the adjustment. Please clarify. Please also consider my other comments on maturity.

This part was revised and we emphasised that the differences were the same regardless the adjustment for maturity. 

• L271-274. Please describe what the Authors mean by physical efficiency.

We replaced “physical efficiency” with “physical fitness” throughout the entire manuscript. Otherwise, we use physical efficiency as a term describing the level of physical fitness. Children who have high level of physical fitness are physically efficient.

---

## [Decision Letter · Decision Letter 1]

4 Feb 2020

Decline of physical activity in early adolescence: A 3-year cohort study

PONE-D-19-20032R1

Dear Dr. Starc,

We are pleased to inform you that your manuscript has been judged scientifically suitable for publication and will be formally accepted for publication once it complies with all outstanding technical requirements.

With kind regards,

Kathryn L. Weston, PhD

Academic Editor

PLOS ONE

Additional Editor Comments (optional):

Please ensure that the minor typographical requests from Reviewer 1 are addressed in the final manuscript submission. 

Reviewers' comments:

Reviewer's Responses to Questions

**Comments to the Author**

1. If the authors have adequately addressed your comments raised in a previous round of review and you feel that this manuscript is now acceptable for publication, you may indicate that here to bypass the “Comments to the Author” section, enter your conflict of interest statement in the “Confidential to Editor” section, and submit your "Accept" recommendation.

Reviewer #1: All comments have been addressed

Reviewer #2: All comments have been addressed

2. Is the manuscript technically sound, and do the data support the conclusions?

Reviewer #1: Yes

Reviewer #2: Yes

3. Has the statistical analysis been performed appropriately and rigorously? 

Reviewer #1: Yes

Reviewer #2: Yes

4. Have the authors made all data underlying the findings in their manuscript fully available?

Reviewer #1: Yes

Reviewer #2: Yes

5. Is the manuscript presented in an intelligible fashion and written in standard English?

Reviewer #1: Yes

Reviewer #2: Yes

6. Review Comments to the Author

Reviewer #1: The authors have done an excellent job in responding to the comments, and the manuscript has been greatly improved. I just have a few minor specific comments below

Abstract

Lines 28, 30, 31, 33: please provide p values

Methods

Line 113 please change N = to n = to be consistent throughout the paper

Results:

Line 263: here you report age as 12.01 ± 1.0. Prior to this you report your mean and SD with no space (i.e. 12.01±1.0). Please check the manuscript for consistency with spaces

Reviewer #2: (No Response)

7. PLOS authors have the option to publish the peer review history of their article (what does this mean?). If published, this will include your full peer review and any attached files.

Reviewer #1: No

Reviewer #2: No

---

## [Editor Report · Acceptance letter]

11 Feb 2020

PONE-D-19-20032R1 

Decline of physical activity in early adolescence: A 3-year cohort study 

Dear Dr. Starc:

I am pleased to inform you that your manuscript has been deemed suitable for publication in PLOS ONE. Congratulations! Your manuscript is now with our production department. 

With kind regards,

on behalf of

Dr. Kathryn L. Weston 

Academic Editor

PLOS ONE